# Numb Suppresses Notch-Dependent Activation of *Enhancer of split* during Lateral Inhibition in the *Drosophila* Embryonic Nervous System

**DOI:** 10.3390/biom14091062

**Published:** 2024-08-26

**Authors:** Elzava Yuslimatin Mujizah, Satoshi Kuwana, Kenjiroo Matsumoto, Takuma Gushiken, Naoki Aoyama, Hiroyuki O. Ishikawa, Takeshi Sasamura, Daiki Umetsu, Mikiko Inaki, Tomoko Yamakawa, Martin Baron, Kenji Matsuno

**Affiliations:** 1Department of Biological Sciences, Graduate School of Science, Osaka University, Toyonaka 560-0043, Japan; elzava@bio.sci.osaka-u.ac.jp (E.Y.M.);; 2Graduate School of Arts and Sciences, University of Tokyo, Meguro 153-8902, Japan; 3Institute for Glyco-Core Research, Gifu University, Gifu 501-1193, Japan; 4Graduate School of Science, Chiba University, Chiba 263-8522, Japan; 5School of Science, Graduate School of Science, University of Hyogo, Ako 678-1297, Japan; inaki@sci.u-hyogo.ac.jp; 6Department of Industrial Engineering, Chemistry, Bioengineering and Environmental Science Course, National Institute of Technology, Ibaraki College, Hitachinaka 312-8508, Japan; 7School of Biological Sciences, Manchester Academic Health Science Centre, University of Manchester, Manchester M13 9PL, UK

**Keywords:** Notch, *numb*, *Enhancer of split*, neurogenesis, lateral inhibition, neurogenic phenotype, *Drosophila*

## Abstract

The role of *Drosophila numb* in regulating Notch signaling and neurogenesis has been extensively studied, with a particular focus on its effects on the peripheral nervous system (PNS). Previous studies based on a single loss-of-function allele of *numb*, *numb^1^*, showed an antineurogenic effect on the peripheral nervous system (PNS), which revealed that the wild-type *numb* suppresses Notch signaling. In the current study, we examined whether this phenotype is consistently observed in loss-of-function mutations of *numb.* Two more *numb* alleles, *numb^EY03840^* and *numb^EY03852^*, were shown to have an antineurogenic phenotype in the PNS. We also found that introducing a wild-type *numb* genomic fragment into *numb^1^* homozygotes rescued their antineurogenic phenotype. These results demonstrated that loss-of-function mutations of *numb* universally induce this phenotype. Many components of Notch signaling are encoded by maternal effect genes, but no maternal effect of *numb* was observed in this study. The antineurogenic phenotype of *numb* was found to be dependent on the *Enhancer of split* (*E*(*spl*)), a downstream gene of Notch signaling. We found that the combination of *E*(*spl*) homozygous and *numb^1^* homozygous suppressed the neurogenic phenotype of the embryonic central nervous system (CNS) associated with the *E*(*spl*) mutation. In the *E(spl)* allele, genes encoding basic helix-loop-helix proteins, such as *m5*, *m6*, *m7*, and m8, remain. Thus, in the *E(spl)* allele, derepression of Notch activity by *numb* mutation can rescue the neurogenic phenotype by increasing the expression of the remaining genes in the *E(spl)* complex. We also uncovered a role for *numb* in regulating neuronal projections. Our results further support an important role for *numb* in the suppression of Notch signaling during embryonic nervous system development.

## 1. Introduction

Cell signaling is critical to a variety of biological processes. The Notch-mediated cell signaling pathway is responsible for a wide range of functions, including cell fate determination and patterning, through direct cell-cell contact throughout the postnatal animal [1,2]. Thus, in humans, aberrant activity of Notch signaling leads to the development of various diseases [3]. The major steps in the cascade leading to the activation of Notch signaling have been elucidated. Notch and the two major ligand families, Delta and Serrate, are single transmembrane proteins [1,2]. Interaction of the extracellular domain of Notch with ligands presented on the surface of adjacent cells causes a conformational change in Notch [4]. These conformational changes make the negative regulatory region of the Notch extracellular domain susceptible to the proteolytic activity of Kuzbanian/ADAM10 [4,5,6,7]. This proteolytic cleavage then leads to another cleavage of the Notch transmembrane domain by γ-secretase, leading to the release of the Notch intracellular domain (NICD) [4,5,6,7,8]. NICD then translocates to the nucleus, where it acts as a co-activator of transcription and activates the expression of target genes such as *Enhancer of split* (*E(spl)*) [5,6,7]. The mechanisms of this signaling cascade and the factors that comprise and regulate the pathway are evolutionarily conserved [4,8].

Various other proteins have been identified as regulators of Notch signaling. These proteins modulate Notch signaling activity through, for example, glycosylation, transport, and degradation to fine-tune Notch activity in ways that are crucial for normal development and homeostasis [4,9,10,11]. The *numb* gene was first identified in *Drosophila* and is known to be involved in the development of the peripheral nervous system (PNS) [12,13], and Numb family proteins are evolutionarily conserved from *Drosophila* to mammals [12,14]. Numb has a conserved phosphotyrosine-binding (PTB) domain near its N-terminal region responsible for its membrane localization [15,16]. In the C-terminal region of Numb, there are regions similar to the Shc PTB domain and an NPF (asparagine-proline-phenylalanine) motif [17]. The NPF motif can interact with the Eps homology (EH) domain in Eps15 [15,17]. These regions play an important role in linking Numb to other proteins involved in asymmetric cell division and Notch signaling inhibition [15,16,17].

The molecular mechanisms by which wild-type *numb* represses Notch signaling have been reported in a variety of developmental contexts [6,16,18]. During asymmetric cell division, Sanpodo (Spdo), a four-transmembrane protein homologous to the actin-related protein tropomodulin, enhances Notch signaling when *numb* is not present [19]. Numb, together with Adaptor protein 1 (AP-1), a protein important for recycling and protein sorting, form a complex that regulates Spdo recycling and prevents Spdo from returning to the cell surface [20,21]. 

In *Drosophila*, embryos homozygous for a loss-of-function mutant of *numb* (*numb^1^*) were previously shown to have fewer peripheral neurons than wild type embryos [12]. This phenotype was considered an “antineurogenic” phenotype, suggesting an overactive Notch signaling [22]. This idea established *numb* as a negative regulator of Notch signaling [16,23]. However, following these analyses, the role of *numb* in the development of the embryonic nervous system has not been studied intensively. Furthermore, these studies on the embryonic nervous system have used almost exclusively the *numb^1^* allele, which did not produce detectable Numb protein on a Western blot [12]. Thus, it was unclear whether suppression of Notch signaling is a universal phenomenon for *numb* loss-of-function alleles. Therefore, to clarify this, we examined the phenotypes of alternative *numb* alleles in the embryonic nervous system. The results showed that various *numb* alleles exhibited the antineurogenic phenotype, suggesting that Notch signaling is generally upregulated in *numb* mutants. Genome rescue experiments also revealed that this antineurogenic phenotype is caused by the loss of function of the *numb* gene. Notch signaling activates a variety of target genes in the embryo [24]. Among these downstream cascades, the loss-of-function mutation of *numb* induced the antineurogenic phenotype mediated through *Notch*-dependent *E(spl)* activation.

## 2. Materials and Methods 

### 2.1. Fly Stocks

The genotypes of the *Drosophila* lines used in this study are as follows: *numb^1^*, a null allele of *numb* (BDSC 4096) [12]; *numb^EY03840^* (DGGR 114573) [25,26] and *numb^EY03852^* (DGGR 114568) [26], both P-element insertion alleles of *numb* gene; *fTRG_25*, a line with a fosmid genomic fragment in which the wild-type *numb* gene is fused in-frame with a DNA fragment encoding superfolder GFP at its 5′-end, *fTRG_25* (FlyFos015836(*pRedFlp-Hgr*) (*numb [41193]::2 XTY1-SGFP-V5-preTEV-BLRP-3XFLAG)dFRT*) (VDRC 318011) [27]; *Df(3)X10,* a *mδ-m3* deletion at the *E(spl)* locus [28]; *ovo^D^ FRT40A* [29]; *Df(2L)N22-3* (DGGR 105876) [30] and *Df(2L)gamma7* (DGGR 108755) [31], two chromosomal deletion lines of *numb*. *Canton-S* was used as a wild type control strain. All flies were raised at 25 °C in a standard *Drosophila* culture medium.

### 2.2. Preparation of Fly Food

Fly food was prepared according to the following recipe for roughly 2400 food vials. A pot was filled with 9 L water, 108 g agar, and 1800 g sugar. We placed the pot on the stove and stirred until the agar and glucose dissolved. We kneaded 1150 g of corn flour, 250 g of corn grit, 720 g of brewer’s yeast, and 324 g of rice bran well with 4.5 L of water until uniform, then added them to a pot. Additionally, 4.5 L of hot water was added. Once it boiled, we reduced the heat and stirred well for 20 min to prevent burning, then turned off the heat and stirred until the mixture cooled to below 75 °C. Subsequently, 54 mL propionic acid, 90 mL butyl p-hydroxybenzoate, 54 mL propionic acid, and 90 mL butyl p-hydroxybenzoate were added. About 7 mL of food was dispensed in each vial.

### 2.3. Generation of numb Maternal and Zygotic Mutant Embryos

FLP-FRT dominant female sterility technique was used to produce *numb* germline clones [32]. *y w hs-FLP; numb^EY03840^*, *FRT40A/CyO hb-lacZ*, or *y w hs-FLP*; *numb^EY03852^*, *FRT40A/CyO hb-lacZ* female flies were mated with *hs-FLP/+*; *ovo^D^*, *FRT40A*/*CyO hb-lacZ* males. These flies were transferred to new vials daily. Late second or early third instar larvae from these crosses were heat-shocked at 37 °C for 1 h, incubated at 18 °C for 2 h, heat-shocked again at 37 °C for 1 h, and incubated at 25 °C until adults emerged. Emerged females were crossed with *y w hs-FLP/+; numb^EY03840^, FRT40A/CyO hb-lacZ* males or *y w hs-FLP/+; numb^EY03852^, FRT40A/CyO hb-lacZ* males to obtain embryos that were *numb* homozygous and lacking maternal contribution, designated as *numb^M/Z^*. The mated flies were placed in collection tubes containing grape agar medium and yeast paste, from which *numb^M/Z^* embryos were collected.

### 2.4. Immunohistochemical Staining

The embryonic stage was determined according to previously described gut morphology [33]. Homozygous mutant embryos were identified based on the absence of balancer chromosomes, *CyO* (*CyO hb-lacZ*), and *TM6B* (*TM6B ubi-GFP*) expressing *lacZ* and *GFP,* respectively. The recovered embryos were decolorized with 50% bleach and fixed in a 1:1 mixture of heptane and 4% paraformaldehyde for 30 min with shaking. The embryos were shaken vigorously in a 1:1 mixture of methanol and heptane to remove the vitelline membrane. Embryos were washed in 100% methanol and stored at −20 °C until immunohistochemical staining. The primary antibodies used for immunostaining were rat anti-embryonic lethal aberration (Elav) (DSHB 7E8A10, 1:500 dilution) [34], mouse anti-Fasciclin II (FasII) (DSHB 1D4, 1:50 dilution) [35], rat anti-Hunchback (Hb) (1:300 dilution) [36], chicken anti-β-galactosidase (β-gal) (Abcam 134435, 1:500 dilution), and rabbit anti-GFP (MBL 598, 1:500 dilution) [37]. Fluorescent Alexa488, Cy3, and Cy5-conjugated secondary antibodies were used at 1:500 dilution (all from Jackson Immunoresearch, West Grove, PA, USA). Embryos were mounted in 50% glycerol and 0.025% n-propyl gallate in PBS.

### 2.5. Scoring the Neurogenic and Antineurogenic Phenotypes of numb Mutants

Each experiment was conducted as at least biological triplicates (three independent crosses) to calculate the percentages of phenotypes in the nervous system of *numb* mutants. We added up the number of embryos obtained from biological replicates.

### 2.6. Confocal Microscope

The embryos were observed using a LSM700 confocal laser microscope (Zeiss). Images were acquired from the apical surface to a depth of 30 µm and constructed as Z-stack photographs (1 μm each). The obtained images were analyzed using LSM Image Browser ZEN 2012 (Zeiss, Jena, Germany) and ImageJ software (Version 13.0.6, NIH, Bethesda, MD, USA) [38].

## 3. Results

### 3.1. Strong Loss-of-Function Mutations in numb Universally Induce an Antineurogenic Phenotype in the PNS of Embryos

The majority of studies on the role of the *numb* gene in embryonic development have been conducted using the null allele called *numb^1^ in Drosophila* [12,13,18,25,39,40]. Pioneering studies reported that embryos homozygous for *numb^1^* show reduced peripheral nervous system (PNS) neurons compared to wild type embryos [12]. This phenotype has been termed the antineurogenic phenotype, suggesting an overactivation of Notch signaling [6,22]. In the present study, we analyzed various *numb* mutant alleles, including *numb^1^, numb^EY03840^*, and *numb^EY03852^*, to confirm the generality of the antineurogenic phenotype with strong loss-of-function mutations in *numb* [12,25,26]. The objective of examining additional *numb* alleles was to verify the universality of the phenotypes and rule out the possibility that the phenotypes in *numb^1^* were due to a second-site mutation or specific allele effect. Ensuring the phenotypes across different alleles confirms that they are due to the loss of *numb* functions but not caused by genetic background. We confirmed that *numb^1^* homozygous embryos exhibit an antineurogenic phenotype, which was previously reported [12]. In wild type and *numb^1^* homozygous embryos, differentiated neurons were detected by immunostaining with an anti-Elav antibody at stage 14 (Figure 1A–C) [34]. Analysis of these embryos resulted in reduced neurons in the PNS compared to the wild type (Figure 1B,C). To investigate whether such an antineurogenic phenotype is universally associated with the *numb* mutant allele, we here examined the PNS phenotype of embryos homozygous for the other *numb* alleles, *numb^EY03840^* and *numb^EY03852^*, known as P-element insertion mutations [25,26]. The results showed that *numb^EY03840^* and *numb^EY03852^* exhibited an antineurogenic phenotype (Figure 1D,E). However, the phenotype of *numb^EY03852^* was mild compared to *numb^1^* and *numb^EY03840^* phenotypes, suggesting that *numb ^EY03852^* is a hypomorphic allele of *numb* (Figure 1C–E). The antineurogenic phenotype of *numb^EY03840^* was as severe as *numb^1^*, a pre-existing null allele of *numb*, suggesting that *numb^EY03840^* is also a null allele of *numb* based on genetic criteria (Figure 1C,D) [12]. On the other hand, heterozygotes of *numb^1^*, *numb^EY03852^,* or *numb^EY03840^* showed normal CNS and PNS, demonstrating that the antineurogenic phenotype is recessive (Figure 1F–H). The finding that *numb^EY03840^* is a null allele of *numb* was further supported by severe antineurogenic phenotype in transheterozygous embryos of *numb^EY03840^* and chromosomal deletion mutants lacking the *numb* locus, *Df(2L)N22-3* (DGGR 105876) or *Df(2L)gamma7* (DGGR 108755) (Figure 1I,J). Transheterozygotes of *numb^1^* and *numb^EY03840^* also showed the antineurogenic phenotype in the PNS, suggesting that the antineurogenic phenotype is caused by mutations in the *numb* gene (Figure 1C,K) [12]. These results collectively demonstrated that loss-of-function mutations in *numb* generally cause an antineurogenic phenotype in the embryonic PNS.

To further confirm this idea, we introduced a genomic fragment containing the wild-type *numb* locus (*FlyFos015836(pRedFlp-Hgr)(numb[41193]::2XTY1-SGFP-V5-preTEV-BLRP-3XFLAG)dFRT*) inserted into the 3rd chromosome, designated as *fTRG_25*, into *numb^1^* homozygotes and investigated whether the antineurogenic phenotype of the PNS is rescued [27]. It was previously shown that a copy of *fTRG_25* is sufficient to rescue the lethality associated with *numb^1^* homozygotes [27]. We here revealed that introducing this genomic fragment into *numb^1^* homozygotes effectively rescued the antineurogenic phenotype of the PNS (Figure 2A–C). Therefore, we conclude that the antineurogenic phenotype in the PNS of *numb* mutants is caused by the loss of function of the *numb* gene.

### 3.2. Maternal numb Does Not Contribute to Notch Signal Downregulation

Various genes encoding components of the Notch signaling pathway are known to have maternal functions in embryonic nervous system development [41,42,43,44,45]. Therefore, it is possible that *numb* may have maternal effects, as discussed in previous studies [12,46,47,48]. To test this possibility, we created homozygous embryos for *numb* mutations without its maternal contribution (*numb^M/Z^* embryos). For this purpose, we employed an FLP/FRT-based method to create germline clones using two *numb* alleles, *numb^EY03840^* or *numb^EY03852^* [29,49]. We observed neurons in wild type and *numb^M/Z^* embryos of *numb^EY03840^* or *numb^EY03852^* using anti-Elav antibody staining (Figure 3A–C). If *numb* has maternal effects, a more severe antineurogenic phenotype should be observed in *numb^M/Z^* embryos than in *numb* homozygous embryos receiving a maternal *numb* gene product supply. However, abnormalities of these *numb^M/Z^* embryos in the nervous system, including the antineurogenic phenotype of the PNS, were nearly equivalent to those of *numb^EY03840^* or *numb^EY03852^* homozygotes (Figure 1C,D and Figure 3B,C). These results suggest that *numb* has no maternal influence on the development of the embryonic nervous system.

### 3.3. The Antineurogenic Phenotype of the numb Mutant Was Dependent on E(spl), a Target Gene of Notch Signaling

The antineurogenic phenotype in *numb* mutants suggested that wild-type *numb* represses Notch signaling [18,50,51,52]. However, it is not known what downstream cascade of Notch signaling is involved in this repression. For example, various Notch signaling target genes, such as *Enhancer of split* (*E(spl)*), *tramtrack* (*ttk*), and *single-minded* (*sim*), are activated before and after neurogenesis in the embryo [53,54]. *E(spl)* is known to play a major role in the development of the CNS and PNS of the embryo [28]. Therefore, wild-type *numb* was postulated to suppress neuron formation in the PNS by upregulating Notch signaling-dependent *E(spl)* activation. The *E(spl)* locus contains multiple genes, such as *m8*, *m7*, *m5*, *m3*, *mβ*, *mγ*, and *mδ*, encoding basic helix-loop-helix transcription factors [55]. *Df(3)X10* mutants have a deletion spanning *mδ*–*m3* at the *E(spl)* locus [28,55]. In *Df(3)X10* homozygotes, the *E(spl)* function is significantly reduced, but the *E(spl)* function remains as the genome contains *m8*, *m7*, *m5*, and *m4* [55]. However, as previously reported, *Df(3)X10* homozygous embryos show severe neuronal hyperplasia, called the neurogenic phenotype, and in this study, this phenotype was also detected by anti-Elav antibody staining (Figure 4C) [28,55]. To analyze the genetic interaction between *numb* and *E(spl*), we combined *numb^1^* and *Df(3)X10* and observed embryonic phenotypes in the PNS and CNS. The double heterozygotes of *numb^1^* and *Df(3)X10* did not show detectable defects in these tissues (Figure 4D). However, the double homozygotes *of numb^1^* and *Df(3)X10* showed no neurogenic phenotype in the CNS or antineurogenic phenotype in the PNS (Figure 4G). The overall structure of the CNS and PNS in these embryos was not much different from that of the wild type (Figure 4A,G). However, potential changes in the number of neurons may not be detected using this experimental procedure. Nevertheless, we revealed that the double homozygotes of *numb^1^* and *Df(3)X10* showed some irregularities in the nervous system structure compared to the wild type (Figure 4A,G). These results revealed an antagonistic genetic interaction between *numb* and *E(spl)*. However, *numb^1^*/+; *Df(3)X10/Df(3)X10* embryos showed a strong neurogenic phenotype comparable to *Df(3)X10/Df(3)X10* (Figure 4C,F). Furthermore, *numb^1^/numb^1^*; *Df(3)X10/+* embryos showed an antineurogenic phenotype comparable to *numb^1^/numb^1^* (Figure 4B,E). Thus, *numb^1^* and *Df(3)X10* did not show dominant suppression of the neurogenic and antineurogenic phenotypes associated with *E(spl)* and *numb* mutations, respectively. However, since *numb* and *E(spl)* mutually suppress neurogenic and antineurogenic phenotypes, our results suggest that *numb* mutants enhance Notch activation, which can upregulate the expression of remaining *E(spl)* genes to compensate for the missing ones and rescue the *Df(3)X10* mutant phenotype (Figure 4H).

### 3.4. Notch Signaling Is Not Upregulated during Lateral Inhibition in the CNS in numb Mutants

Our results so far indicate that *numb* suppresses the neurogenic phenotype of the *E(spl)* mutant in the CNS, suggesting that *numb* mutations upregulate Notch signaling in the CNS and the PNS (Figure 4G,H). Therefore, we examined whether lateral inhibition is enhanced in the CNS of *numb* mutants at embryonic stage 9 when neuroblast differentiation begins. Neuroblast identity is determined by several transcription factors, such as *Hunchback* (*hb*) [56,57]. Hence, *hb* is a marker of neuroblasts whose number is regulated by Notch signaling via lateral inhibition [58,59].

To observe neuroblasts, we detected Hunchback (Hb) protein, a marker of neuroblasts, by immunostaining with anti-Hb antibody in embryos at stage 9 (Figure 5A,B). The results showed that neuroblasts in the neuroectoderm of *numb^1^* homozygous embryos were largely comparable to the wild type (Figure 5A,B). To quantitatively analyze the numbers of neuroblasts, we counted the number of Hb-positive nuclei in these embryos (Figure 5C). The average number of neuroblasts did not significantly differ between *numb^1^* homozygous and wild type embryos (Figure 5C). This result is consistent with the observation that the CNS observed by anti-Elav antibody staining appeared normal in *numb^1^* homozygotes (Figure 1B).

### 3.5. numb Mutations Impaired Neuronal Projection in the PNS

We examined the neuronal projections of *numb* mutants to determine if the lack of *numb* function causes abnormalities other than an antineurogenic phenotype in the development of the nervous system. The intersegmental nerve (ISN), crucial for the *Drosophila* PNS, is formed by pioneering motor neurons extending axons from the CNS and joining sensory axons. Cell adhesion molecules, particularly Fasciclin II (FasII), play a vital role in axon fasciculation by mediating homophilic interactions [60]. FasII is a member of the immunoglobulin superfamily that predominantly localizes to neurites [60,61]. Thus, FasII is a suitable marker for observing neurite projection. In this study, we used anti-FasII antibody staining to analyze the structure of neurite projections in the embryonic CNS and PNS. The results showed that *numb^1^, numb^EY03840^*, or *numb^EY03852^* homozygotes had no detectable abnormalities in FasII-positive axons in the CNS (Figure 6B–D).

While no abnormalities were detected in FasII-positive axons in the CNS, the PNS showed significant disorganization (Figure 7B–D). In *numb^1^, numb^EY03840^*, or *numb^EY03852^* homozygotes, FasII-positive neurite projection of the ISN axons became disorganized and occasionally shorter than the wild type (Figure 7A–D). To quantitatively analyze these defects, we obtained the percentages of the abnormally shorter ISN axons (Table 1). The shorter ISNs were observed more frequently in the *numb^1^, numb^EY03840^*, or *numb^EY03852^* homozygotes compared with the wild type (Table 1). This indicates that *numb* is essential for proper axonal projections in the PNS. At this stage, however, the mechanism regarding how the pathways of these axons are disrupted in *numb* mutants remains unclear.

## 4. Discussion

The importance of numb in the downregulation of Notch signaling has been studied in a variety of species [16,62]. In *Drosophila*, only the *numb^1^* allele has been used in early studies that revealed an antineurogenic phenotype in *numb^1^* homozygous PNS [12]. Because of this, it was not clear whether such antineurogenic phenotypes were universally associated with loss-of-function mutations in *numb.* In this study, we addressed this question using two other *numb* alleles, *numb^EY03840^* and *numb^EY03852^*. The results revealed that the loss-of-function mutants of *numb* universally exhibit an antineurogenic phenotype in the PNS (Figure 1C–E). To further confirm this finding, we examined whether the antineurogenic phenotype of *numb^1^* homozygotes could be rescued by the introduction of a genomic fragment containing the wild-type *numb* locus [27]. In embryos homozygous for *numb^1^* and carrying this genomic fragment, the antineurogenic phenotype in the PNS was suppressed (Figure 2C). Furthermore, as reported previously, these embryos survived to adulthood, indicating that the recessive lethality associated with the *numb^1^* mutant was rescued by this genomic fragment [27]. It is unclear what anomalies in the *numb^1^* mutant are responsible for the rescue of such recessive lethality. However, based on these results, we conclude that *numb* loss-of-function mutations universally induce an antineurogenic phenotype in the PNS [22].

It has been suggested that the maternal *numb* may compensate for the deletion of the zygotic *numb* and provide sufficient functions for a part of normal development during early embryogenesis [12,46,47,48]. In this study, we tested this possibility using the *numb^EY03840^* allele, which was genetically suggested to be a null mutant. Previous studies have shown that *numb* has a maternal-specific alternative-splicing product [12,63,64]. Since *numb^EY03840^* has a P-element inserted in an exon common to maternal and zygotic transcripts, this insertion likely disrupts both maternal and zygotic *numb* function in *numb^EY03840^* mutation, suggesting that *numb^EY03840^* is null maternally and zygotically [63,65,66]. In the present study, embryos of zygotes and maternal mutants of *numb^EY03840^* exhibited an antineurogenic phenotype comparable to that of zygotic homozygotes of *numb^EY03840^* (Figure 1D and Figure 3B). These results suggest that *numb* has no maternal effect, or if it does, it is negligible.

During embryonic development, Notch signaling exerts its function through the activation of various downstream target genes such as *E(spl)*, *ttk*, and *sim* [53]. In this study, we found that *numb* and *E(spl)* mutually suppress antineurogenic and neurogenic phenotypes, respectively (Figure 4G). *Df(3)X10* is a deletion mutant of *E(spl)* that lacks several genes in the *E(spl)* complex, such as *m3*, *mβ*, *mγ*, and *mδ*. These genes encode basic helix-loop-helix proteins and are known to be involved in lateral inhibition through Notch signaling [67,68]. *Df(3)X10* homozygotes showed a strong neurogenic phenotype, but as previously reported, other genes of the *E(spl)* complex, such as *m5, m6, m7*, and *m8*, remain in *Df(3)X10* [55]. Wild-type *numb* suppresses *E(spl)* activation by downregulating Notch. Thus, in *Df(3)X10* homozygotes, derepression of Notch activity by *numb* mutation can rescue the neurogenic phenotype by increasing the expression of the remaining genes in the *E(spl)* complex.

Importantly, *numb^1^* suppressed the neurogenic phenotype in the CNS of *Df(3)X10*. This suggests that wild-type *numb* suppresses Notch signaling-mediated lateral inhibition in the neuroectoderm. Therefore, we considered the potential role of *numb* in the neuroectoderm and examined the expression of the *hb* gene, a marker for neuroblasts, in *numb^1^* homozygotes. However, the number of neuroblasts in *numb^1^* homozygotes was equivalent to that of wild type (Figure 5C). This result agrees with the previous finding that the number of neuroblasts in the CNS was not affected in the *numb^1^* homozygote [69]. Thus, in wild type embryos where the transcription of the *E(spl)* complex was activated at the normal level through Notch signaling, the absence of *numb* did not result in the hyperactivation of lateral inhibition in the neuroectoderm.

Since cell-fate determination to neurons is impaired in *numb* mutants, we speculated that such defects might lead to abnormalities other than neuronal differentiation. To test this possibility, we compared the neurite projection in *numb* mutants and wild type embryos. *In Drosophila,* FasII is often used as a marker for neurite projections of motor neurons [70,71]. In particular, anti-FasII antibody staining is useful for analyzing neurite structure. We found that the neurites of PNS motoneurons in *numb* mutants are disorganized and occasionally shorter (Figure 7B–D). These results suggest that wild-type *numb* is required for proper projection of the motor neurons. Numb interacts with endocytic components and affects the trafficking and localization of guidance receptors during axon guidance [52]. Thus, the loss of *numb* functions may impair axonal responses to guidance cues, leading to the observed ISN abnormalities. However, the defect in neurite projection may be due to the lack of some neurons in the *numb^1^* mutant. Therefore, future studies need to elucidate the molecular and cellular mechanisms by which *numb* regulates neurite projection.

## Figures and Tables

**Figure 1 biomolecules-14-01062-f001:**
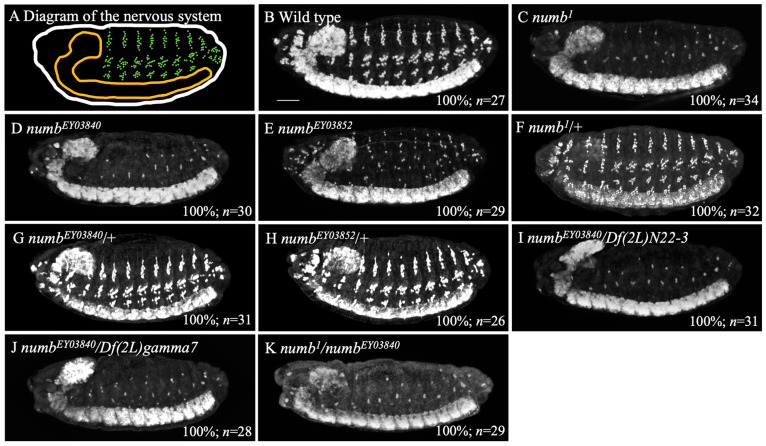
*numb* loss-of-function alleles exhibited an antineurogenic phenotype in the PNS. A schematic diagram of the CNS (orange) and PNS (green) in a *Drosophila* embryo at stage 14 (**A**). Embryos of wild type (**B**), *numb^1^* homozygote (**C**), *numb^EY03840^* homozygote (**D**), *numb^EY03852^* homozygote (**E**), *numb^1^* heterozygote (**F**), *numb^EY03840^* heterozygote (**G**), *numb^EY03852^* heterozygote (**H**), *numb^EY03840^/Df(2L)N22-3* transheterozygote (**I**), *numb^EY03840^/Df(2L)gamma7* transheterozygote (**J**), and *numb^1^/numb^EY03840^* transheterozygote (**K**) were stained with an anti-Elav antibody. Lateral views of the embryos are shown. In these embryos, neurons were shown in white. The percentage of embryos showing the phenotype presented is shown in % at the bottom right. The number of embryos examined is indicated by “*n*”. The scale bar represents 25 µm.

**Figure 2 biomolecules-14-01062-f002:**
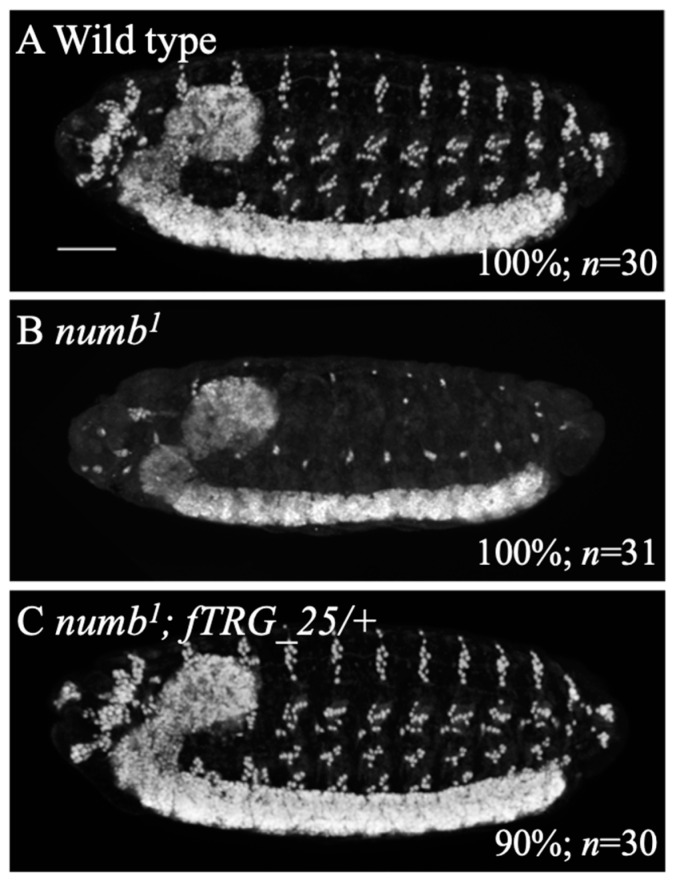
The antineurogenic phenotype of *numb^1^* homozygous was rescued by introducing a genomic fragment containing the wild-type *numb* locus. Embryos of wild type (**A**), *numb^1^* homozygote (**B**), and *numb^1^; fTRG_25/+* (**C**) were stained with an anti-Elav antibody. *fTRG_25* carries the wild-type genomic fragment of the *numb* gene. Lateral views of the embryos are shown. In these embryos, neurons were observed in white. The percentage of embryos showing the phenotype presented is shown in % at the bottom right. In C, 10% of embryos showed an antineurogenic phenotype. The number of embryos examined is indicated by “*n*”. The scale bar represents 25 µm.

**Figure 3 biomolecules-14-01062-f003:**
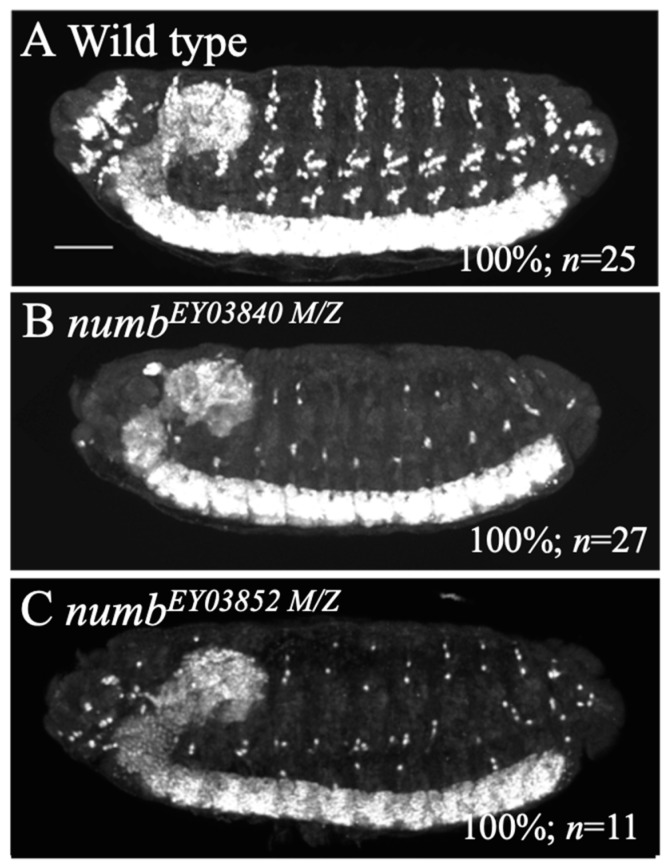
Embryos homozygous for *numb* mutants and lacking its maternal contribution showed an antineurogenic phenotype comparable to zygotic homozygotes for the *numb* alleles in the PNS. Embryos of wild type (**A**), *numb^EY03840^* homozygote lacking maternal contribution of *numb* (*numb^EY03840 M/Z^*) (**B**), and *numb^EY03852^* homozygote lacking maternal contribution of *numb* (*numb^EY03852 M/Z^*) (**C**) were stained with an anti-Elav antibody. Lateral views of the embryos are shown. In these embryos, neurons were observed in white. The percentage of embryos showing the phenotype presented is shown in % at the bottom right. The number of embryos examined is indicated by “*n*”. The scale bar represents 25 µm.

**Figure 4 biomolecules-14-01062-f004:**
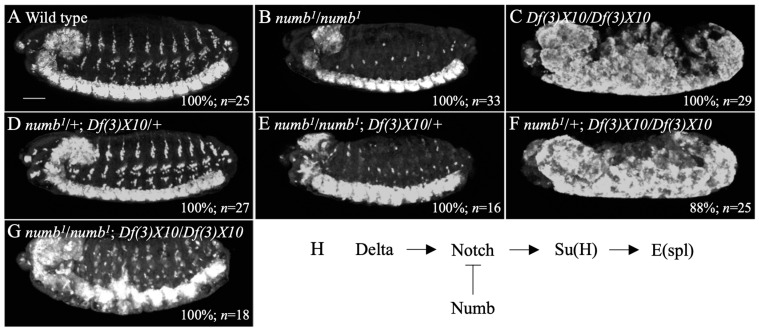
*numb* and *E(spl)* mutually suppressed the neurogenic and antineurogenic phenotypes, respectively. Embryos of wild type (**A**), *numb^1^* homozygote (**B**), *Df(3)X10* homozygote (**C**), *numb^1^*/+; *Df(3)X10/+* (**D**), *numb^1^/numb^1^; Df(3)X10/+* (**E**), *numb^1^*/+; *Df(3)X10/Df(3)X10* (**F**), and *numb^1^/numb^1^; Df(3)X10/Df(3)X10* (**G**) were stained with an anti-Elav antibody. *Df(3)*X10 is a deletion mutant of the *E(spl)* complex. Lateral views of the embryos are shown. In these embryos, neurons were shown in white. The percentage of embryos showing the phenotype presented is shown in % at the bottom right. The number of embryos examined is indicated by “*n*”. The scale bar represents 25 µm. A schematic diagram of Notch signaling analyzed here is shown (**H**).

**Figure 5 biomolecules-14-01062-f005:**
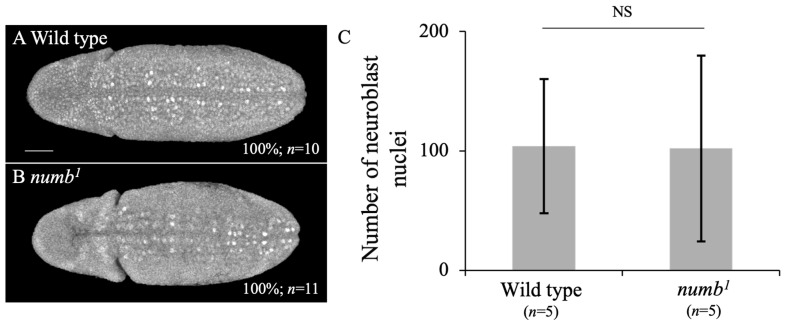
*numb^1^* homozygotes showed a similar number of neuroblasts. Wild type (**A**) and *numb^1^* homozygous (**B**) embryos were stained with an anti-Hb antibody. Ventral views of the embryos are shown. Neuroblasts were observed to be white in these embryos. The percentage of embryos showing the presented phenotype is in % at the bottom right. The number of embryos examined is indicated by “*n*”. The scale bar represents 25 μm. The average numbers of Hb-positive nuclei are shown (**C**). Standard deviations are shown as bars at the top of each bar. The *t*-test showed no significant differences between the wild type and *numb^1^* embryos. The numbers of embryos examined are shown at the bottom of each bar.

**Figure 6 biomolecules-14-01062-f006:**
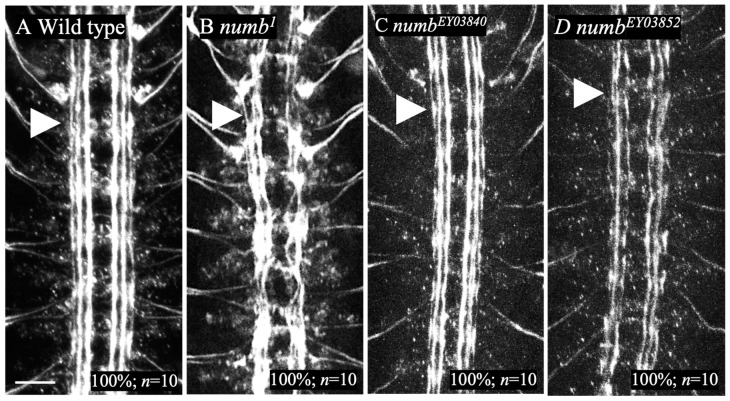
Axons of motor neurons in the CNS were normal in *numb* homozygote mutants. Axons of CNS motoneurons stained with an anti-FasII antibody in wild type (**A**), *numb^1^* homozygous (**B**), *numb^EY03840^* homozygous (**C**), and *numb^EY03852^* homozygous (**D**) embryos. Ventral views of the embryos are shown. In these embryos, axons were shown in white. The percentage of embryos showing the presented phenotype is in the lower right corner as %. The number of embryos examined is indicated by “*n*”. The scale bar represents 25 µm. Axon tracts are indicated by white arrowheads.

**Figure 7 biomolecules-14-01062-f007:**
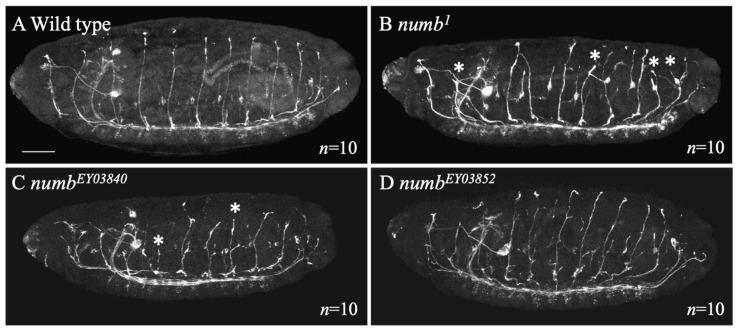
Axons of motor neurons in the PNS were disorganized in *numb* homozygotes. Axons of PNS motoneurons stained with an anti-FasII antibody in wild type (**A**), *numb^1^* homozygous (**B**), *numb^EY03840^* homozygous (**C**), and *numb^EY03852^* homozygous (**D**) embryos. Lateral views of the embryos are shown. In these embryos, axons were shown in white. The number of embryos examined is indicated by “*n*”. The scale bar represents 25 μm. Abnormally short ISNs are indicated by white asterisks.

**Table 1 biomolecules-14-01062-t001:** The percentage of abnormally short axons in wild type and *numb* mutants.

Genotype	% of Abnormally Short Axons
Wild type	0% (*n* = 10)
*numb^1^*	38% (*n* = 10)
*numb^EY03840^*	25% (*n* = 10)
*numb^EY03852^*	4% (*n* = 10)

## Data Availability

Data are contained within the article.

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
