# Peer review of "Numb Suppresses Notch-Dependent Activation of Enhancer of split during Lateral Inhibition in the Drosophila Embryonic Nervous System"

_biomolecules, 2024, doi:10.3390/biom14091062_

Round 1
Reviewer 1 Report
Comments and Suggestions for Authors
The manuscript by Mujizah et al. characterizes neurogenesis phenotypes of several mutations in the developmentally important Drosophila gene, numb. A hallmark of numb loss-of-function, namely the antineurogenic phenotype of the embryonic peripheral nervous system, is demonstrated for two previously uncharacterized numb mutants. One of these mutants is likely a second null allele of numb as homozygotes completely phenocopy the effects of the original numb1 null mutation. Rescue of the antineurogenic phenotype of numb1 by a genomic fragment containing the wildtype numb gene is demonstrated, and strong evidence is provided that numb’s role in embryonic PNS development is due to its zygotic rather than its maternal contribution. Somewhat weaker evidence is provided that numb also functions in CNS neurogenesis and perhaps also nerve organization. Remarkably, homozygous null mutations of numb are shown to substantially rescue the severe CNS neurogenic defects of the Enhancer-of-split mutation Df(3)X10). Evidence is also presented that numb mutants have reduced numbers of neuroblasts in developing embryos and that their motor neuron axons form aberrant nerves.
Major Comments:
Most of the results in the manuscript are relatively straightforward and augment our understanding of numb function. The manuscript does, however, have several minor weaknesses. One of these is its relative lack of quantitation. All of the data is presented in the format of a representative image accompanied by the number of animals examined and the percentage of animals with the phenotype shown. This is particularly problematic in cases where the differences or similarities between phenotypes are not so obvious, such as in the demonstration of suppression of the Df(3)X10 phenotype by numb1 shown in Fig. 4. The images in 4A and 4G are said to have the same (normal) phenotype with respect to neurogenesis. However, the distribution of neurons clearly differs in the two cases and it is far from clear that the cell numbers are comparable, particularly in the CNS where the labeling in G is both broader and brighter than in A. Some quantitation of the actual numbers of neurons generated in the two genotypes would provide more conclusive evidence for similarity. Similarly, the claim that central neuroblast numbers are reduced by numb1 mutation (Fig. 5) would benefit from quantification of the results. The differences in staining shown in Fig. 5 are not unambiguously the result of changes in cell number. They could instead be the result of reduced staining intensity and excluding this possibility and providing a quantitative accounting of neuroblast numbers would be more convincing.
The demonstration of ISN defects in numb1 mutants is also not particularly convincing and merits a clearer demonstration, particularly as the question of what defects occur in the formation of the intersegmental nerve in the absence of (many or all) sensory axons is an interesting one. As it stands, the description of the ISN motoneuron axon defects in numb mutants is too incomplete to be useful. It’s not clear what is meant when the ISNs in numb mutants are described as "disordered," and the images in Fig. 7 do not make it any clearer. Branching patterns of the ISN are very well known and to illustrate what deficits occur, the authors should provide a schematic of the portion of the ISN examined showing the branching patterns in wildtype and mutant animals, with example images. The authors should also provide some background to help the reader understand what is known about the formation of the ISN (which is pioneered by both motor and sensory axons) and what therefore might be going wrong in the mutants. Is the defect likely to be a central one? Or is it a consequence of the reduced numbers of sensory neurons in the periphery? While it’s true as the authors state that the mechanisms remain to be determined, the Discussion should contain at least some informed speculation as to the possible causes of the ISN defects and how they might relate to the antineurogenic phenotype.
Minor Comments
ll. 25-6: Although the demonstration of antineurogenic phenotypes for the two new numb mutant alleles is useful information, the motivation for examining them is not really clear. The manuscript repeatedly states that the P-element alleles were tested to determine whether the antineurogenic phenotypes observed with numb1 were “universal.” But why test these alleles and, more importantly, why is it important to know if the phenotype is universal? Is there any reason to doubt that numb1 is a null mutation? Is there a possibility, for example, that the antineurogenic phenotype ascribed to numb1 is actually due to a “second site” mutation? Or is there a problem with interpreting numb’s function if some (e.g. hypomorphic) alleles don’t have an antineurogenic phenotype? The authors need to explain why examination of the additional numb alleles is useful.
Fig. 1: This figure and others showing the antineurogenic phenotypes have a single arrowhead and asterisk to refer to the PNS and CNS neurons, respectively. Although most readers are likely to know that these labels refer to broader sets of cells than those indicated there’s room for confusion. A reader might not know that all of the clusters (and not just the single cluster labeled with an arrowhead) represent peripheral neurons. Perhaps a schematic (or more extensively labeled figure panel) in Fig. 1 making it clear what is CNS and what is PNS would be helpful.
l. 182 Where is the fosmid inserted in the genome? That is, on which chromosome?
ll. 187-9: The wording here is confusing as it sounds like rescue of both lethality and the antineurogenic phenotype were previously reported.
Fig. 2C: The figure indicates that 10% of numb1 mutant embryos were not rescued by the fosmid. Was there no rescue or partial rescue? Please explain.
l. 206: “homozygous for numb” should probably read “homozygous for numb mutations.”
Fig. 5: As noted in the general comments, it's actually not entirely clear from the images in A and B that the number of NBs is reduced. It looks like the level of anti-Hb labeling may simply be reduced. The addition of images at higher power and/or intensity might be useful for showing the reduction, as would actual cell counts quantifying the reduction.
ll. 286-8: The explanation here doesn’t seem very compelling. Why should upregulation of Notch signaling result in only a "slight decrease" in neuron number? In other words, why would the effects of Notch upregulation on NB number be so slight when it is large enough to completely counter the dramatic overproduction of neurons caused by E(spl) mutation (i.e. by Df(3)X10)?
ll. 369-72: No data is actually presented that directly demonstrates the downregulation of Notch (and that such downregulation is responsible for the suppression of the E(spl) phenotype), so this conclusion should be tempered. Also, the authors should comment on the suppression of the antineurogenic phenotype of numb mutants by Df(3)X10. What explains this? A figure with a schematic showing the known interactions of the various players (numb, Notch, E(spl)) in the PNS and CNS and those interactions hypothesized on the basis of the data presented would be extremely helpful. How these pathways give rise to the results could then be indicated graphically.
Author Response
Major Comment:
1. Most of the results in the manuscript are relatively straightforward and augment our understanding of numb function. The manuscript does, however, have several minor weaknesses. One of these is its relative lack of quantitation. All of the data is presented in the format of a representative image accompanied by the number of animals examined and the percentage of animals with the phenotype shown. This is particularly problematic in cases where the differences or similarities between phenotypes are not so obvious, such as in the demonstration of suppression of the Df(3)X10 phenotype by numb1 shown in Fig. 4. The images in 4A and 4G are said to have the same (normal) phenotype concerning neurogenesis. However, the distribution of neurons clearly differs in the two cases and it is far from clear that the cell numbers are comparable, particularly in the CNS where the labeling in G is both broader and brighter than in A. Some quantitation of the actual numbers of neurons generated in the two genotypes would provide more conclusive evidence for similarity.
Response 1:
Thank you very much for your valuable comments. Although we agree with REVIEWER I, we found that it would not be easy to quantify the number of neurons in Figure 1 A and G, because each embryo has too many neurons to count. However, we agree with this point raised by the reviewer. Thus, to address it, we modified the sentence to explain that the number of neurons could differ among them, although the overall structure of their nervous system appeared to be similar.
Modified sentence: page 7, line 267-272
The overall structure of the CNS and PNS in these embryos was not much different from the wild type, although potential changes in the number of neurons could not be detected using this experimental procedure (Fig. 4 A and G). Nevertheless, the double homozygotes of numb1 and Df(3)X10 showed some irregularities in the nervous system structure compared to the wild type (Fig. 4 A and G).
2. Similarly, the claim that central neuroblast numbers are reduced by numb1 mutation (Fig. 5) would benefit from quantification of the results. The differences in staining shown in Fig. 5 are not unambiguously the result of changes in cell number. They could instead be the result of reduced staining intensity and excluding this possibility and providing a quantitative accounting of neuroblast numbers would be more convincing.
Response 2:
The reviewer requested a quantitative analysis to reveal the effect of numb1 on the number of neuroblasts. As pointed out by the reviewer, we found that staining with Hb antibody, representing the number of neuroblasts, was not clear enough to count the number of them. Therefore, we conducted the anti-Hb staining again and counted the number of Hb-positive nuclei. To quantitatively present these results, we show their average numbers as graphs in Figure 5C. However, in the new set of experiments, we did not detect a significant difference in the number of neuroblasts between the wild type and numb1 mutant. Therefore, in the revised manuscript, we withdraw our previous conclusion that the number of neuroblasts decreased in the numb1 mutant. Instead, we concluded that neuroblasts did not decrease in the numb1 mutant, compared with the wild type. To describe these results, we modified the sentences as below. We also added the following sentence to explain the graphs in Figure 5C.
Modified sentences: page 8, line 303-310
The results showed that neuroblasts in the neuroectoderm of numb1 homozygous embryos were largely compared to the wild type (Fig.5 A and B). To quantitatively analyze the numbers of neuroblasts, we counted the number of Hb-positive nuclei in these embryos (Fig.5 C). The average number of neuroblasts did not significantly differ between numb1 homozygous and wild-type embryos (Fig.5 C). This result is consistent with the observation that the CNS observed by anti-Elav antibody staining appeared normal in numb1 homozygotes (Fig.1 B).
Modified sentences: page 11, line 408-413
However, the number of neuroblasts in numb1 homozygotes otherwise wild type was equivalent to that of wild type (Fig. 5C). This result agrees with the previous finding that the number of neuroblasts was not affected in numb1 homozygote [71]. Thus, in wild-type embryos where E(spl) transcription was activated at the normal level through Notch signaling, the absence of numb did not result in the hyperactivation of lateral inhibition in the neuroectoderm.
Added sentence: page 9, line 317-320
The average numbers of Hb-positive nuclei are shown (C). Standard deviations are shown as bars at the top of each bar. T-test showed no significant difference between wild type and numb1. The numbers of embryos examined are shown at the bottom of each bar.
3. The demonstration of ISN defects in numb1 mutants is also not particularly convincing and merits a clearer demonstration, particularly as the question of what defects occur in the formation of the intersegmental nerve in the absence of (many or all) sensory axons is an interesting one. As it stands, the description of the ISN motoneuron axon defects in numb mutants is too incomplete to be useful. It’s not clear what is meant when the ISNs in numb mutants are described as "disordered," and the images in Fig. 7 do not make it any clearer. Branching patterns of the ISN are very well known and to illustrate what deficits occur, the authors should provide a schematic of the portion of the ISN examined showing the branching patterns in wildtype and mutant animals, with example images.
Response 3:
Thank you very much for another important comment for improving the clarity of our manuscript. The percentages of ISN motoneurons with abnormally shorter axons were scored and shown in Table 1. To describe the table, we added the following sentences:
Added sentence: page 10, line 345-348
To quantitatively analyze these defects, we obtained the percentages of the abnormally shorter ISN axons (Table 1). The shorter ISN were observed more frequently in numb1, numbEY03840, or numbEY03852 homozygote, compared with the wild type (Table 1).
Added sentence: Table 1 page 10
The percentages of the abnormally shorter ISN in wild type and homozygotes of numb1, numbEY03840, or numbEY03852 are shown. The numbers of embryos examined are shown in parentheses.
4. The authors should also provide some background to help the reader understand what is known about the formation of the ISN (which is pioneered by both motor and sensory axons) and what therefore might be going wrong in the mutants. Is the defect likely to be a central one? Or is it a consequence of the reduced numbers of sensory neurons in the periphery? While it’s true as the authors state that the mechanisms remain to be determined, the Discussion should contain at least some informed speculation as to the possible causes of the ISN defects and how they might relate to the antineurogenic phenotype.
Response 4:
Thank you very much for another constructive suggestion. We found this suggestion valuable as it will improve the clarity of discussion. To discuss the point, we added the following sentences to Results and Discussion.
Added sentence: page 9, line 324-328
The intersegmental nerve (ISN), crucial for Drosophila's peripheral nervous system (PNS), is formed by pioneering motor neurons extending axons from the CNS and joining sensory axons. Cell adhesion molecules, particularly Fasciclin II (FasII), play a vital role in axon fasciculation by mediating homophilic interactions [61].
Added sentence: page 12, line 420-423
Numb interacts with endocytic components and affects the trafficking and localization of guidance receptors during axon guidance [53]. Thus, the loss of numb functions may impair axonal responses to guidance cues, leading to the observed ISN abnormalities.
Minor Comments
1. II. 25-6: Although the demonstration of antineurogenic phenotypes for the two new numb mutant alleles is useful information, the motivation for examining them is not really clear. The manuscript repeatedly states that the P-element alleles were tested to determine whether the antineurogenic phenotypes observed with numb1 were “universal.” But why test these alleles and, more importantly, why is it important to know if the phenotype is universal? Is there any reason to doubt that numb1 is a null mutation? Is there a possibility, for example, that the antineurogenic phenotype ascribed to numb1 is actually due to a “second site” mutation? Or is there a problem with interpreting numb’s function if some (e.g. hypomorphic) alleles don’t have an antineurogenic phenotype? The authors need to explain why examination of the additional numb alleles is useful.
Response 1:
To address the comment from the reviewer, we added the following sentence:
Added sentence: page 4, line 166-170
The objective of examining additional numb alleles was to verify the universality of the phenotypes and rule out the possibility that the phenotypes in numb1 are due to a second-site mutation or specific allele effect. Ensuring the phenotypes across different alleles confirms that they are due to the loss of numb functions but not caused by genetic background.
2. Fig. 1: This figure and others showing the antineurogenic phenotypes have a single arrowhead and asterisk to refer to the PNS and CNS neurons, respectively. Although most readers are likely to know that these labels refer to broader sets of cells than those indicated there’s room for confusion. A reader might not know that all of the clusters (and not just the single cluster labeled with an arrowhead) represent peripheral neurons. Perhaps a schematic (or more extensively labeled figure panel) in Fig. 1 making it clear what is CNS and what is PNS would be helpful.
Response 2:
Thank you very much for the valuable suggestion. We added a schematic diagram of the CNS and PNS in an embryo in Fig. 1A.
Added sentence: page 5, line 196
A schematic diagram of the CNS (yellow) and PNS (green) in Drosophila embryo at stage 14 (A).
3. I. 182 Where is the fosmid inserted in the genome? That is, on which chromosome?
Response 3:
It is on the third chromosome. To describe this information, we added the following sentence.
Modified sentence: page 5, line 207
To further confirm this idea, we introduced a genomic fragment containing the wild-type numb locus (FlyFos015836(pRedFlp-Hgr)(numb[41193]::2XTY1-SGFP-V5-preTEV-BLRP-3XFLAG)dFRT) into the 3rd chromosome, designated as fTRG_25, into numb1 homozygotes and investigated whether the antineurogenic phenotype of the PNS is rescued [28].
4. II. 187-9: The wording here is confusing as it sounds like rescue of both lethality and the antineurogenic phenotype were previously reported.
Response 4:
We agree with this comment. As pointed out, we modified the sentences as below.
Modified sentence: page 5, line 209-212
It was previously shown that a copy of fTRG_25 is sufficient to rescue the lethality associated with numb1 homozygotes [28]. We here revealed that introducing this genomic fragment into numb1 homozygotes effectively rescued the antineurogenic phenotype of the PNS (Fig.2 A-C).
5. Fig. 2C: The figure indicates that 10% of numb1 mutant embryos were not rescued by the fosmid. Was there no rescue or partial rescue? Please explain.
Response 5:
It was “no rescue.” To describe this, we added the following sentence:
Added sentence: page 6, line 221
In C, 10% of embryos showed normal CNS and PNS.
6. I. 206: “homozygous for numb” should probably read “homozygous for numb mutations.”
Response 6:
We corrected the sentence as suggested.
7. Fig. 5: As noted in the general comments, it's actually not entirely clear from the images in A and B that the number of NBs is reduced. It looks like the level of anti-Hb labeling may simply be reduced. The addition of images at higher power and/or intensity might be useful for showing the reduction, as would actual cell counts quantifying the reduction.
Response 7:
Thank you very much for this instructive comment. As suggested, we conducted experiments again to quantitatively analyze the number of neuroblasts (see major comment 2). In these analyses, we did not detect a significant difference in the number of neuroblasts between the wild type and numb1 mutant. As our response to the major comment 2, we took back our previous conclusion that the number of neuroblasts was reduced in the numb1 mutant. Instead, we showed new results that neuroblasts did not decrease in the numb1 mutant, compared with the wild type. Please see our responses to major comment 2 to know how we modified sentences.
8. II. 286-8: The explanation here doesn’t seem very compelling. Why should upregulation of Notch signaling result in only a "slight decrease" in neuron number? In other words, why would the effects of Notch upregulation on NB number be so slight when it is large enough to completely counter the dramatic overproduction of neurons caused by E(spl) mutation (i.e. by Df(3)X10)?
Response 8:
Thank you very much for your thoughtful comment. As pointed out by the reviewer, it is hard to explain what occurs in the cascade of Notch signaling under this condition. Especially, feedback events may need to be considered. However, to clarify our findings, we added the following sentence.
Added sentence: page 11, line 410-412
Thus, in wild type embryos where the transcription of E(spl) complex was activated at the normal level through Notch signaling, the absence of numb did not result in the hyperactivation of lateral inhibition in the neuroectoderm.
9. II. 369-72: No data is actually presented that directly demonstrates the downregulation of Notch (and that such downregulation is responsible for the suppression of the E(spl) phenotype), so this conclusion should be tempered. Also, the authors should comment on the suppression of the antineurogenic phenotype of numb mutants by Df(3)X10. What explains this? A figure with a schematic showing the known interactions of the various players (numb, Notch, E(spl)) in the PNS and CNS and those interactions hypothesized on the basis of the data presented would be extremely helpful. How these pathways give rise to the results could then be indicated graphically.
Response 9:
Thank you very much. As suggested, we added the diagram of the Notch pathway discussed in this paper to Figure 4 H.
Reviewer 2 Report
Comments and Suggestions for Authors
In this paper, Mujizah et al., examined the role of numb during neurogenic process. They showed that loss of numb causes antineurogenic phenotype, decreased number of HB+ neuroblast, and defective ISN axon projection. The data is convincing and the images are nicely taken. Manuscript is also well written. However, I have a few concerns about the claims they made in this manuscript (please refer to my review point 1 and 2). In addition, I would suggest two additional experiments for general interests (point 3 and 4).
1. The authors are sought to test whether the antineurogenic phenotype is related to numb mutants. They observed that numbEY03852 exhibited milder antineurogenic effects and concluded that numbEY03852 is a hypomorphic allele. However, the authors should conduct qPCR or western blot to confirm the nature of this allele. If this allele is also a null mutant rather than hypomorphic, that will suggest the antineurogenic phenotype is NOT universally associated with numb mutation. Transhets of numbEY03852 and the deficiencies may also provide hints to the nature of this allele – if it is indeed hypomorphic, transhets may have fewer functional Numb than numbEY03852 homozygotes, but a bit more than the nulls, therefore we may expect a phenotype somewhere in between.
2. The authors claimed that the near WT phenotype observed in Figure 4G is due to “numb mutants enhance Notch activation, which can upregulate the expression of remaining E(spl) genes to compensate for the missing ones and rescue the Df(3)X10 mutant phenotype.” However, current data does not support this claim. The authors could perform qPCR to confirm the elevation of the expression of the remaining E(spl) genes.
3. It is interesting that the increased lateral inhibition in numb1 mutant leads to fewer HB+ neuroblasts. As the authors may know, HB is a temporal TF that is transiently expressed in neuroblasts. The reason of fewer HB+ NBs could be due to a change of HB expression time window. A delay of NB differentiation could also lead to fewer HB+ NBs at stage 9. It will be interesting to assay a later stage to see the expression of HB or other TF (like Kr) downstream of HB.
4. Given that there are fewer PNS neurons in numb1 mutant, the defects of PNS axon projection could be simply due to the loss of neurons. This might fit with the observation that some ISN axon bundles are shorter. The authors could examine the number of PNS neurons (or just count the number of the sensory neurons that project through ISN). The number of neurons (or even the identity of neurons) in each hemisegment might correlate to the ISN axon phenotype.
Author Response
1. The authors are sought to test whether the antineurogenic phenotype is related to numb mutants. They observed that numbEY03852 exhibited milder antineurogenic effects and concluded that numbEY03852 is a hypomorphic allele. However, the authors should conduct qPCR or western blot to confirm the nature of this allele. If this allele is also a null mutant rather than hypomorphic, that will suggest the antineurogenic phenotype is NOT universally associated with numb mutation. Transhets of numbEY03852 and the deficiencies may also provide hints to the nature of this allele – if it is indeed hypomorphic, transhets may have fewer functional Numb than numbEY03852 homozygotes, but a bit more than the nulls, therefore we may expect a phenotype somewhere in between.
Response 1:
Thank you very much for your valuable comment. The numbEY03852 allele is a P-element insertion allele and the insertion site was determined previously. A reference was cited to provide this information. As pointed out by the reviewer, the phenotype of numbEY03852 was slightly weaker than those of the null alleles of numb (numb1 and numbEY03840). To examine whether numbEY03852 is a hypomorphic allele, the reviewer requested to show the phenotype of transheterozygote between numbEY03852 and a deletion uncovering numb locus. However, we have conducted similar experiments investigating the phenotypes of numbEY03852/numb1. Since numb1 is a null allele, we can perform these genetic tests using this allele instead of deletions uncovering numb locus. As predicted by the reviewer, our preliminary data showed that numbEY03852/numb1 seemed to have an intermediate phenotype between numbEY03852 homozygote and numb1 homozygote. However, this difference was too subtle to draw a concrete conclusion. Therefore, although we respect this comment, we decided not to include this result in our revision to prevent introducing ambiguous data.
2. The authors claimed that the near WT phenotype observed in Figure 4G is due to “numb mutants enhance Notch activation, which can upregulate the expression of remaining E(spl) genes to compensate for the missing ones and rescue the Df(3)X10 mutant” However, current data does not support this claim. The authors could perform qPCR to confirm the elevation of the expression of the remaining E(spl) genes.
Response 2:
Thank you very much for the instructive suggestion. Unfortunately, in the E(spl) locus of the Df(3)X10 mutant genome, multiple genes encode helix-loop-helix transcription factors, such as m8, m7, m5, and m4 still exist. Thus, to perform qPCR, we need to detect the transcripts of these genes. We found that the preparation of these experiments will take a long time. Therefore, we believe these experiments are worth trying, we did not include them in our revision.
3. It is interesting that the increased lateral inhibition in numb1 mutant leads to fewer HB+ neuroblasts. As the authors may know, HB is a temporal TF that is transiently expressed in neuroblasts. The reason of fewer HB+ NBs could be due to a change of HB expression time window. A delay of NB differentiation could also lead to fewer HB+ NBs at stage 9. It will be interesting to assay a later stage to see the expression of HB or other TF (like Kr) downstream of HB.
Response 3:
Thank you very much for your valuable suggestion. Although we agree with this comment, our experiment involving anti-Hb antibody staining at stage 9 is a typical approach for detecting Notch signaling during lateral inhibition. Therefore, we believe that the suggested experiment involving anti-Kr antibody staining would not add additional insight to the manuscript. Hence, we did not perform this experiment in our revision, although we are grateful to the reviewer for the instructive comment.
4. Given that there are fewer PNS neurons in numb1 mutant, the defects of PNS axon projection could be simply due to the loss of neurons. This might fit with the observation that some ISN axon bundles are shorter. The authors could examine the number of PNS neurons (or just count the number of the sensory neurons that project through ISN). The number of neurons (or even the identity of neurons) in each hemisegment might correlate to the ISN axon phenotype.
Response 4:
We found this comment very important for improving our manuscript. As suggested by the reviewer, we attempted to count the number of neurons. However, since each ISN is composed of multiple neurons connected, it was difficult to identify each neuron and count their number. Therefore, instead of counting the neuron’s numbers, we counted abnormally short ISNs and calculated the percentages of them in wild type, numb1, numbEY03840, and numbEY05852. These results are shown in Table 1. To describe Table 1, we added the sentence below to our revision. In addition, in previous pictures of ISNs in Fig. 7, it was difficult to see the short ISNs due to their high magnification. Therefore, we replaced them with new pictures with lower magnification, which show the short ISNs more clearly. As predicted by the reviewer, we found some correlation between the percentages of abnormally short ISNs and the severity of the antineurogenic phenotype. Therefore, defective projection of ISNs may be caused by missing neurons, as pointed out by the reviewer. However, although we respect the comment, the defective projection of ISNs can be explained by other mechanisms, for example, the defects in cell-fate decision and abnormality in pathfinding machinery. Therefore, it would be hard to interpret the data obtained from the suggested experiment. Thus, we did not include further discussion regarding this correlation.
Added sentence: page 10, line 345-348
To quantitatively analyze these defects, we obtained the percentages of the abnormally shorter ISN axons (Table 1). The shorter ISN were observed more frequently in numb1, numbEY03840, or numbEY03852 homozygote, compared with the wild type (Table 1).
Added sentence: Table 1 page 10
The percentages of the abnormally shorter ISN in wild type and homozygotes of numb1, numbEY03840, or numbEY03852 are shown. The numbers of embryos examined are shown in parentheses.
Reviewer 3 Report
Comments and Suggestions for Authors
Abnormal activity of Notch signaling can result in the onset of various diseases in humans. This is becasue the Notch-mediated cell signaling pathway plays a crucial role in numerous functions, such as cell fate determination and patterning, facilitated through direct cell-to-cell contact throughout the postnatal life of an organism. Previous studies based on a single loss-of-function allele of numb, known as numb1, demonstrated an antineurogenic effect on the peripheral nervous system (PNS). These findings revealed that the wild-type numb suppresses Notch signaling. In their submitted manuscript, Elzava et al. demonstrate that loss-of-function mutations of numb universally induce an antineurogenic phenotype. They found that this antineurogenic effect of numb depends on the Enhancer of split [E(spl)] and uncovered a novel role for numb in regulating neuronal projections, which is interesting. Building on these findings, the combination of E(spl) homozygous and numb1 homozygous suppressed the neurogenic phenotype of the embryonic central nervous system (CNS) associated with the E(spl) mutation. This suggests that numb plays a role in lateral inhibition within the CNS. To me, this work represents a step forward in the understanding of numb1 and Notch signaling. I have the following questions/comments.
1. Figure 1: What is the phenotype of numb1, numbEY03852 , and numbEY03840 Heterozygotes? Is the phenotype of these Heterozygotes the same as wild type (A)?
2. Figure 2, Are the results (90%) representative of three independent experiments? For the statistics, it is good enough to present as such if the value is 100%. If the statistics is not 100%, please do three independent experiments and show the values of each independent experiments. The same query applies to Figure 4.
3. Figure 4, the authors only use the numb1, please use one more allele, numbEY03852 or numbEY03840, to perform the further demonstration.
4. Figure 5, Can E(spl) potentially influence the phenotype resulting from a reduced number of neuroblasts in numb1?
5. Please provide the complete DNA sequence of the fTRG_25 fragment if available, or alternatively, specify the start and end positions of the genomic fragment encompassing the wild-type numb locus in the methodology section.
6. Please write the recipe and ingredients of Drosophila culture medium in detail in the method.
7. The method section requires substantial improvement to enhance its clarity and reproducibility for other researchers.
8. Some modifications/corrections in the manuscript may be needed.
Comments on the Quality of English Language
Author Response
1. Figure 1: What is the phenotype of numb1, numbEY03852, and numbEY03840 heterozygotes? Is the phenotype of these Heterozygotes the same as wild type (A)?
Response 1:
Thank you very much for your instructive comment. We conducted these experiments and found that numb1, numbEY03852, and numbEY03840 heterozygotes showed normal CNS and PNS. We showed these results in Fig. 1 F-H. To describe them, we added the following sentence.
Added sentence: page 4, line 183-185
On the other hand, heterozygotes of numb1, numbEY03852, or numbEY03840 showed normal CNS and PNS, demonstrating that the antineurogenic phenotype is recessive (Fig. 1 F-H).
2. Figure 2, Are the results (90%) representative of three independent experiments? For the statistics, it is good enough to present as such if the value is 100%. If the statistics is not 100%, please do three independent experiments and show the values of each independent experiments. The same query applies to Figure 4.
Response 2:
Thank you very much for your valuable comment. Experiments were conducted with at least biological triplicates (three independent crosses). To calculate the percentages of each phenotype, we summed up the number of embryos obtained from biological replicates. Since this is a standard method to score phenotypes, we did not modify this procedure in our revision. However, as pointed out by the reviewer, we did not mention it in our manuscript. Thus, we added the procedure to calculate the percentages of the phenotype, as follows.
Added sentence: page 3, line 146-149
Scoring the neurogenic and antineurogenic phenotypes of numb mutants
Each experiment was conducted as at least biological triplicates (three independent crosses) to calculate the percentages of phenotypes in the nervous system of numb mutants. We summed up the number of embryos obtained from biological replicates.
3. Figure 4, the authors only use the numb1, please use one more allele, numbEY03852 or numbEY03840, to perform the further demonstration.
Response 3:
In the first part of this manuscript, we showed that numb1 behaves like a typical null allele of numb. Therefore, we used numb1 as the representative null allele of numb. Hence, although we respect this suggestion, we believe that adding other numb alleles in Figure 4 would not give important insight.
4. Figure 5, Can E(spl) potentially influence the phenotype resulting from a reduced number of neuroblasts in numb1?
Response 4:
We agree with this comment. However, to answer this question, a substantial amount of work will be required, which would be beyond the scope of this manuscript. Therefore, we did not include this experiment in our revision.
5. Please provide the complete DNA sequence of the fTRG_25 fragment if available, or alternatively, specify the start and end positions of the genomic fragment encompassing the wild-type numb locus in the methodology section.
Response 5:
fTRG_25 was published as a thesis. Thus, the reference of it was included in our manuscript.
6. Please write the recipe and ingredients of Drosophila culture medium in detail in the method.
Response 6:
We added the following sentence to describe the ingredients of the Drosophila culture medium.
Added sentence: page 3, line 106-115
Preparation of fly food
Fly food was prepared according to the following recipe for roughly 2,400 food vials. A pot was filled with 9 l water, 108 g agar, and 1,800 g sugar. Place the pot on the stove and stir until the agar and glucose dissolve. Knead 1,150 g corn flour, 250 g corn grit, 720 g brewer's yeast, and 324 g rice bran well with 4.5 l water until uniform, then add them to the pot. Additionally, 4.5 l of hot water was added. Once it boils, reduce the heat and stir well for 20 minutes to prevent burning. Turn off the heat and stir until the mixture cools to below 75ËšC. Subsequently, 54 ml propionic acid, 90 ml butyl p-hydroxybenzoate, 54 ml propionic acid, and 90 ml butyl p-hydroxybenzoate were added. About 7 ml of food was dispensed in each vial.
7. The method section requires substantial improvement to enhance its clarity and reproducibility for other researchers.
Response 7:
Thank you very much for your valuable comment. As our response to this comment, we added the procedure to score the phenotypes (comment 2) and the ingredients of Drosophila culture medium (comment 5).
8. Some modifications/corrections in the manuscript may be needed.
Response 8:
As suggested, we checked and modified the manuscript again.
Round 2
Reviewer 1 Report
Comments and Suggestions for Authors
The authors did a nice job responding to my comments. I have no further concerns.